# Patterns and Dominant Driving Factors of Carbon Storage Changes in the Qinghai–Tibet Plateau under Multiple Land Use Change Scenarios

Huihui Zhao [1] , Caifeng Yang [2], Miao Lu [3,*], Longhao Wang [4] and Bing Guo [4,*]

[1] Research Institute of Aerospace Information, Chinese Academy of Sciences, Beijing 100101, China; zhaohh@radi.ac.cn
[2] Beijing WANYUN Science & Technology Development Co., Ltd., Beijing 100081, China; yangcaifeng@bjwanyun.com
[3] State Key Laboratory of Efficient Utilization of Arid and Semi-Arid Arable Land in Northern China—Institute of Agricultural Resources and Regional Planning, Chinese Academy of Agricultural Sciences, Beijing 100081, China
[4] School of Civil Engineering and Geomatics, Shandong University of Technology, Zibo 255000, China; 23507020856@stumail.sdut.edu.cn
* Correspondence: lumiao@caas.cn (M.L.); guobing@sdut.edu.cn (B.G.)

**Abstract:** Revealing the spatial–temporal evolution of carbon storage and its driving mechanisms in the Qinghai–Tibet Plateau could provide support for decision making in the protection of regional ecosystems and the achievement of regional dual-carbon goals. In this study, the spatial–temporal evolution of carbon storage in the Qinghai–Tibet Plateau was analyzed under various scenarios using PLUS-InVEST and a gravity center model, and the driving mechanisms of carbon storage were clarified with Geodetector. The results are as follows: (1) During 2000–2020, the areas of coniferous forest, evergreen broad-leaved forest, closed shrub, temperate shrub desert, multi-tree grassland, and grassland showed an increasing trend, while the areas of deciduous broad-leaved forest and mixed forest showed a decreasing trend. (2) During 2030–2060, there was a decreasing trend in the total carbon storage of the Qinghai–Tibet Plateau under three different scenarios. (3) During 2030–2060, the area of the Qinghai–Tibet Plateau was mostly represented by carbon balance (56%), while the areas of carbon sources and carbon sinks showed a scattered distribution. (4) The precipitation and topographic factors with a q value of 0.888 played a dominant role in affecting the spatio-temporal variations in carbon storage in the Qinghai–Tibet Plateau. (5) In future ecological protection and restoration efforts, more high-quality farmlands should be protected and constructed, which could contribute to the achievement of dual-carbon goals. In addition, the hydrothermal conditions should be improved to aid the carbon cycle process in the Qinghai–Tibet Plateau.

**Keywords:** carbon storage; dominant factor; spatio-temporal changes; carbon sink; InVEST model



## 1. Introduction

Human society has been significantly developed through the increase in the global economic level and the progress of science and technology. However, at the same time, the problem of climate change has become increasingly prominent. The climate situation in the world has been deteriorating since the industrial revolution. Unreasonable land use patterns, such as excessive consumption of fossil fuels, over-reclamation of forests and grasslands, and overgrazing, have led to an increase in greenhouse gas emissions. Climate-related extreme weather phenomena, such as droughts, high temperatures, and floods, are also increasing, which seriously affect our living environment and safety [1,2]. Today, controlling the rise in global temperature and achieving carbon neutrality have become the common goals of all countries in the world to address climate change. Land use change is one of the important causes of the increase in carbon emissions, and its contribution is

second only to fossil fuel combustion [3]. Land use change directly or indirectly affects the fixed carbon function of ecosystems. It is a key driving factor for the surface carbon cycle and one of the core areas of global climate change research [4].

In terms of land use change prediction, a number of Land Use and Land Cover Change (LUCC) prediction models used both in China and worldwide have been reported. These models can be divided into two categories: quantitative prediction models and spatial prediction models. Lai et al. (2019) used the System Dynamics (SD) model to explore the ecological and environmental problems of the water resources system in Fuxian County and obtained a set of more robust control strategies [5]. Wang et al. (2022) used the System Dynamics model to design six different development scenarios and obtain a sustainable development plan to achieve an optimal balance between the ecological environment and social economy in the region by 2030 [6]. Lafond et al. (2018) used the Markov process to explore the dynamic characteristics of poverty changes in low-income groups and demonstrated the application of this method in Peru [7]. Zhu et al. (2018) analyzed the influences of urban spatial development on the ecological environment in the Yangtze River Delta based on the Conversion of Land Use and its Effects at Small region extent (CLUE-S) model and the Markov chain model [8]. Li et al. (2020) predicted the land use scenario of Fangchenggang City in the next 25 years using a Markov–Cellular Automata (CA) combined model and socio-economic data and provided analytical support for local land planning [9]. Koo et al. (2019) used the Geographically Weighted Regression (GWR) model to predict housing prices in Seoul [10]. Hu et al. (2020) used geostatistical methods for the stability assessment of water source land in Shanghai [11]. The Integrated Valuation of Ecosystem Services and Tradeoffs (InVEST) model has been widely used in the field of ecosystem carbon storage research, which includes marine, terrestrial, and freshwater models. Among them, remote sensing data have been widely used in the carbon module, which provides a convenient and rapid method to estimate carbon stocks. Huang et al. (2020) used the hydrological module in the InVEST model to investigate the impacts of different economic activities on water resources in the Kashi River Basin of Xinjiang and put forward corresponding ecological compensation policy recommendations [12]. Wang et al. (2021) used the land use change module in the InVEST model to simulate the future land use change in the North China Plain from 2017 to 2027 and explored the influences of related factors on land use change [13]. The Qinghai–Tibet Plateau ecosystem not only has many ecological functions, such as soil and water conservation, biodiversity conservation, and regional climate regulation, but also plays an important role in the ecological carbon cycle [14]. Few studies have been conducted to predict and reveal the change patterns of carbon storage in the future (2030–2060), which is important for the achievement of regional dual-carbon goals. However, the carbon and nitrogen cycle process of the Qinghai–Tibet Plateau ecosystem has undergone a significant change in recent years due to the influence of climate change and the increase in human activity intensity, which in turn has affected its ecological functions. Although some studies have been conducted on the dynamic changes in the carbon and nitrogen cycle in the Qinghai–Tibet Plateau, the carbon cycle process and its key factors of its influence on various ecosystems in the Qinghai–Tibet Plateau have not been fully revealed and clarified, which has restricted the promotion of ecological function maintenance strategies and adaptive management in the region [15]. Therefore, exploring the relationships between driving factors and ecosystem carbon storage in the Qinghai–Tibet Plateau is necessary to provide scientific support for improving the regional carbon sequestration capacity in the area.

In this paper, the PLUS model was utilized to predict the land use change data under a natural development scenario, a cultivated land protection scenario, and an ecological protection scenario of the Qinghai–Tibet Plateau from 2030 to 2060, and then the InVEST model was applied to obtain the carbon storage in different periods and different scenarios. The objective of this paper was to analyze the spatial–temporal evolutions of carbon storage in the Qinghai–Tibet Plateau under various scenarios and clarify the driving mechanisms of carbon storage changes using Geodetector.

## 2. Materials and Methods

### 2.1. Study Area

The Qinghai–Tibet Plateau is located in western China (Figure 1), covering Tibet, Qinghai, Sichuan, Yunnan, and other provinces. It is located between 73°31′ and 103°04′ E and 26°02′ and 40°14′ N, with a total area of approximately $2.2 \times 10^7$ km². It starts from the Qilian Mountains and the Kunlun Mountains in the east and extends southwestward to the Himalayas with Qinghai Province and Gansu Province as the edges. It reaches Nepal and India to the south and crosses the Kunlun Mountains to the Junggar Basin and Xinjiang to the north [16]. The Qinghai–Tibet Plateau has a continental plateau climate and a monsoon climate, with obvious terrain and altitude distribution [14]. The plateau is dry and rainless, and the temperature changes noticeably. It is cold in winter and cool in summer, with precipitation mainly occurring in summer. The temperature gradually decreases and the precipitation gradually increases with an increase in altitude [13]. The average altitude is higher than 4000 m and the terrain is complex; it is a mountainous region with many steep mountains. Due to the high altitude and the perennial nature of glaciers, many have developed on the Qinghai–Tibet Plateau, thus forming a typical glacier landform [10]. The vegetation types are relatively singular, with mainly grassland, marsh, and alpine plants, of which grassland accounts for more than 80% of the total area and mountain grassland and cultivated land are staggered [5]. The main soil types are alpine meadow soil, swamp soil, and peat soil.

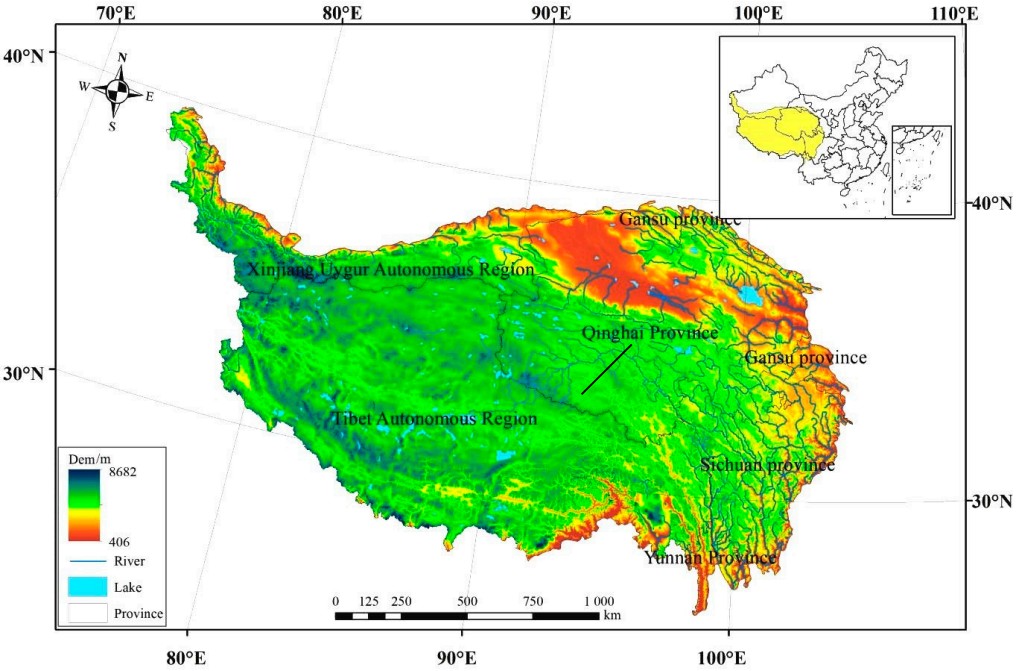

**Figure 1.** Location of the study area.

### 2.2. Data Source and Preprocessing

Land use data from 2000, 2005, 2010, 2015, and 2020 with an overall accuracy of 88.6% were obtained from the MCD12Q1 dataset (https://search.earthdata.nasa.gov/). The original data format was .hdf, and the spatial resolution was 500 m. The land use data were resampled to 1000 m and projected to the Krasovsky_1940_Albers coordinate system by MODIS Reprojection Tool (MRT) (United States Geological Survey, Reston, VA, USA). The 2000–2020 Gross Domestic Product data were obtained from the Resource and Environment Science and Data Center (http://www.resdc.cn), with a spatial resolution of 1 km. The 2000–2020 population data were obtained from LandScan (https://landscan.ornl.gov), with a spatial resolution of 1 km. The climate station data were obtained from the China Meteorological Data Service Center (http://data.cma.cn/). The Kriging interpolation

method in the ArcGIS 10.7 (Environmental Systems Research Institute, Inc., Red Lands, CA, USA) geostatistical analysis module was applied to obtain the precipitation and temperature datasets with a spatial resolution of 1000 m and an overall accuracy of 90.6%. The Digital Elevation Model data were derived from the SRTM2 dataset with a spatial resolution of 90 m (http://www.Gscloud.cn). Then, the slope data were extracted using the slope tool in ArcGIS10.7 (Environmental Systems Research Institute, Inc., Red Lands, CA, USA). The data on national highways, provincial highways, and railways in China were acquired from the data of the Resource Science Center (http://www.resdc.cn) with a spatial resolution of 1 km. The distances from national highways, provincial highways, and railways were calculated using ArcGIS 10.7(Environmental Systems Research Institute, Inc., Red Lands, CA, USA) Euclidean distance (Table 1).

**Table 1.** Data introduction.

| Data Type | Data Name | Data Sources |
|---|---|---|
| Fundamental data | Administrative boundary and prefecture boundary of Qinghai–Tibet Plateau Administrative boundary and prefecture boundary of Qinghai–Tibet Plateau | Resource and Environment Science and Data Center |
| | Land use types for 2000, 2005, 2010, 2015, 2020 | USGS |
| Driving data | DEM Metrological data, rivers, lakes, railways, provincial roads, and national roads | Resource and Environment Science and Data Center |
| Restriction data | Cultivated land distribution data | Resource and Environment Science and Data Center |
| | Distribution data of ecological red line area | Natural Resources Authority |
| Carbon density table | —— | [14–16] |

### 2.3. Methods

The methodology adopted in the current investigation was as follows (Figure 2):

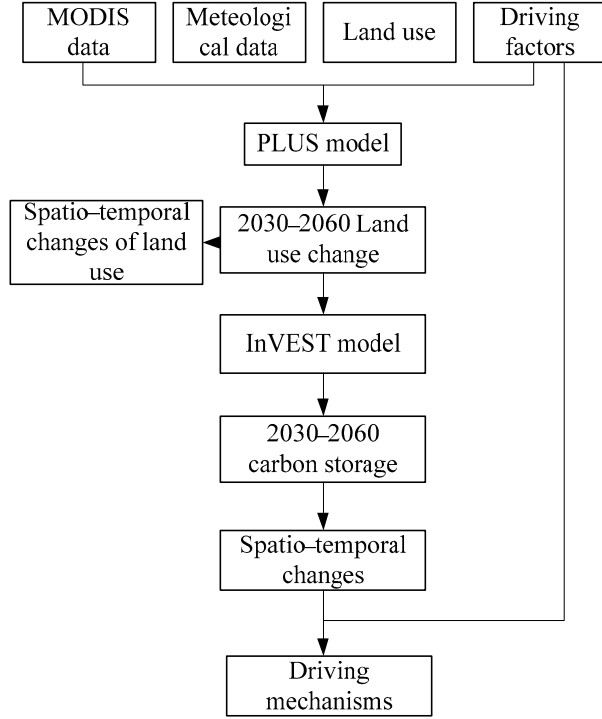

**Figure 2.** Flowchart of the methodology adopted in the current investigation.

### 2.3.1. PLUS Model

The patch-generating land use simulation (PLUS) model is a land use simulation model based on raster data developed by Wuhan University. It can simulate landscape patterns with higher accuracy. The PLUS model can be coupled with the land expansion analysis strategy (LEAS) and the CA model based on multi-type random patch seeds (CARS) to simulate the distribution pattern of land use in various scenarios [17].

(1) Analysis strategy of land expansion

$$P_{i,k(x)}^{d} = \frac{\sum\limits_{n=1}^{M} I(h_{\mathrm{n}}(x) = d)}{M} \tag{1}$$

In the formula, $P_{i,k(x)}^{d}$ refers to the development probability of class $k$ in grid $i$; $x$ refers to a vector composed of multiple driving factors; $h_{\mathrm{n}}(x)$ refers to the prediction type of the $n$th decision tree of vector $x$; when the value of d is 1, it indicates that other land use types are transformed into the k-type land use, and a value of 0 indicates other transformations; $M$ refers to the total number of decision trees; and $I$ refers to the indicator function of the decision tree.

(2) CA model based on multi-accumulated random patch seeds. The module is divided into 2 parts:

(a) Feedback mechanism of macro demand and local competition.

$$OP_{i,k}^{d=1,t} = P_{i,k}^{d=t} \Omega_{i,k}^{t} D_{k}^{t} \tag{2}$$

In the formula, $P_{i,k(x)}^{d}$ refers to the development probability of land type $k$ in grid $i$; $D_{k}^{t}$ refers to the influence of future demand of land type $k$; $\Omega_{i,k}^{t}$ and z is the neighborhood weight of grid $i$.

(b) Decreases in the multi-class random patch seed threshold.

$$OP_{i,k}^{d=1,t} = \begin{cases} P_{i,k}^{d=1}(r\mu k)D_{k}^{t}, & \Omega_{i,k}^{t} = 0 \text{ and } r < P_{i,k}^{d=1} \\ P_{i,k}^{d=1}\Omega_{i,k}^{t}D_{k}^{t}, & \text{others} \end{cases} \tag{3}$$

In the formula, the value of $r$ is [0, 1], and $\mu_{k}$ is the threshold value of the new land class $k$ patch.

### 2.3.2. InVEST Model

The carbon module of the InVEST model has been adopted, in which the carbon storage of the ecosystem is composed of aboveground biomass (Cabove), belowground biomass carbon storage (Cbelow), soil carbon storage (Csoil), and dead organic matter carbon storage [18]. The calculation formula is as follows:

$$C_i = C_{i,above} + C_{i,below} + C_{i,soil} + C_{i,dead} \tag{4}$$

$$C_{total} = \sum_{i=1}^{n} C_i \times S_i \tag{5}$$

In the formula, $C_i$ represents the total carbon density of $i$-type land, $C_{total}$ represents the total carbon storage, $C_{i,above}$ represents the aboveground biomass carbon density of $i$-type land, $C_{i,below}$ represents the underground biomass carbon density of $i$-type land, $C_{i,soil}$ represents the soil organic carbon density of $i$-type land, $C_{i,dead}$ represents the dead organic carbon density of $i$-type land, and $S_i$ represents the total area of $i$-type land.

### 2.3.3. Geodetector

The geographical detector model is used to detect the spatial heterogeneity of carbon stocks. It makes it possible to reveal the explanatory power of certain factors on carbon

stocks. This explanatory power is measured using a q value, and the value range of q is [0, 1]. The larger the value, the stronger the explanatory power of the factor on carbon storage, and vice versa [19–23]. The expression is as follows:

$$q = 1 - \frac{\sum_{h=1}^{L} N_h \sigma_h^2}{N\sigma^2} = 1 - \frac{\text{SSW}}{\text{SST}} \tag{6}$$

$$\text{SSW} = \sum_{h=1}^{L} N_h \sigma_h^2 \tag{7}$$

$$\text{SST} = N\sigma^2 \tag{8}$$

where $h = 1, \ldots, L$ refers to the classification of variable Y or factor X; $N_h$ and $N$ represent the number of layers, $h$, and the number of units, respectively; $\sigma_h^2$ represents the variance in the Y value of class $h$; and $\sigma^2$ represents the variance in the Y value. SSW represents the sum of intra-layer variance, and SST represents the total variance.

## 3. Results

### 3.1. Spatial and Temporal Characteristics of Land Use Change

From 2000 to 2020, grassland was the most widely distributed land use type in the Qinghai–Tibet Plateau, accounting for 51.69%, 52.52%, and 52.29% of the study area, respectively, followed by bare or low-coverage grassland (Table 2). The areas of the remaining land types were ranked as follows: multi-tree grassland > mixed forest > coniferous forest > temperate shrub desert > deciduous broad-leaved forest > evergreen broad-leaved forest > closed shrub. The ratio of closed shrub area is the lowest. From 2000 to 2020, the land use types showed different change trends. The areas of arboreal forest, evergreen broad-leaved forest, closed shrub, temperate shrub desert, multi-tree grassland, grassland, and other land use increased, while the areas of deciduous broad-leaved forest, mixed forest, and bare or low-coverage grassland decreased. Compared with 2000, the areas of coniferous forest, evergreen broad-leaved forest, closed shrub, temperate shrub desert, multi-tree grassland, grassland, and other land use increased by 2568 km$^2$, 25 km$^2$, 7 km$^2$, 9297 km$^2$, 2737 km$^2$, 152.00 × 10$^2$ km$^2$, and 9481 km$^2$ by 2020, accounting for 7.28%, 0.93%, 7.29%, 61.17%, 4.59%, 1.16%, and 13.33%, respectively. Meanwhile, the areas of deciduous broad-leaved forest, mixed forest, and bare or low-cover grassland decreased by 1687 km$^2$, 470 km$^2$, and 371.58 × 10$^2$ km$^2$, accounting for 21.06%, 1.03%, and 3.75%, respectively.

**Table 2.** Land use structure change in Qinghai–Tibet Plateau from 2000 to 2020.

| Land Use Type | 2000 | | 2010 | | 2020 | |
|---|---|---|---|---|---|---|
| | Area/km$^2$ | Proportion/% | Area/km$^2$ | Proportion/% | Area/km$^2$ | Proportion/% |
| Coniferous forest | 35,275 | 1.39 | 35,203 | 1.39 | 37,843 | 1.49 |
| Evergreen broad-leaved forest | 2679 | 0.11 | 2608 | 0.10 | 2704 | 0.11 |
| Deciduous Broad-leaved forest | 8012 | 0.32 | 7087 | 0.28 | 6325 | 0.25 |
| Mixed forest | 45,645 | 1.80 | 44,512 | 1.75 | 45,175 | 1.78 |
| Closed shrub | 96 | 0.00 | 125 | 0.00 | 103 | 0.00 |
| Temperate shrub desert | 15,199 | 0.60 | 18,439 | 0.73 | 24,496 | 0.96 |
| Multi-tree grassland | 59,576 | 2.34 | 65,327 | 2.57 | 62,313 | 2.45 |
| Grassland | 1,313,885 | 51.69 | 1,334,946 | 52.52 | 1,329,085 | 52.29 |
| Bare or low-coverage Grassland | 990,195 | 38.96 | 961,751 | 37.84 | 953,037 | 37.50 |
| Other land use | 71,116 | 2.80 | 71,680 | 2.82 | 80,597 | 3.17 |
| Total | 2,541,678 | 100.00 | 2,541,678 | 100.00 | 2,541,678 | 100.00 |

In order to further analyze the land transfer patterns between different land use types during 2000–2020, this study calculated the area of transferred land use and then produced a transferred land use chord diagram using Origin. Figure 3a shows that from 2000 to 2010, bare or low-coverage grassland was the largest land type and then the area decreased by $284.44 \times 102$ km$^2$. The main land use types were grassland and other land use, accounting for 83.11% and 15.84% of the bare or low-coverage grassland area, respectively. The land type with the largest transferred area was grassland, with an increase of $210.61 \times 102$ km$^2$. Grassland was mainly transferred from bare or low-coverage grassland, other land use, multi-tree grassland, and temperate shrub desert, accounting for 78.78%, 6.35%, 6.27%, and 6.14% of the transferred area of grassland, respectively. The increased grassland was mostly transferred from bare or low-coverage grassland. Closed shrub, evergreen broad-leaved forest, and coniferous forest were all transferred to a small degree.

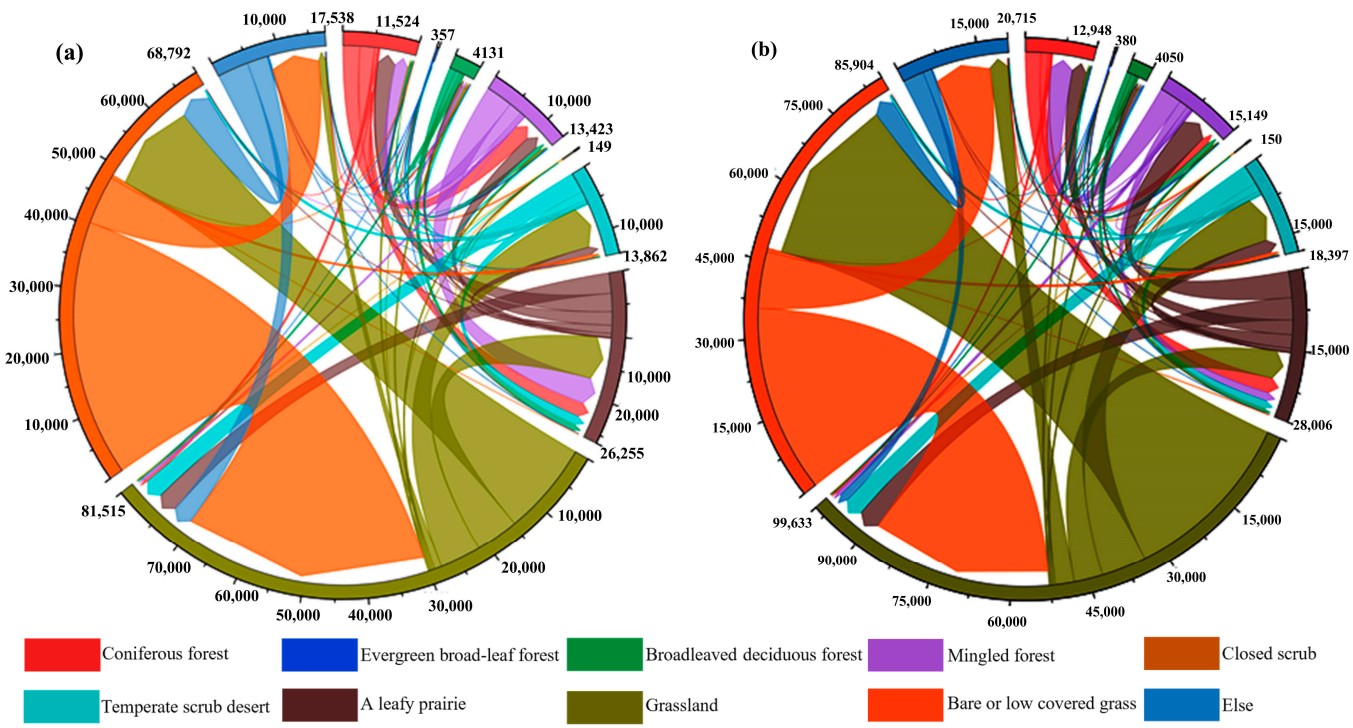

**Figure 3.** Transfer patterns between different land types in the Qinghai–Tibet Plateau from 2000 to 2020: (**a**) 2000–2010; (**b**) 2010–2020.

Figure 3b shows that from 2010 to 2020, the land type with the largest transferred area was bare and low-coverage grassland, with a decrease in area of 8714 km$^2$. The main transferred areas were from grassland and other land use, accounting for 75.65% and 22.91% of the transferred area of bare or low-coverage grassland, respectively. The areas of bare or low-coverage grassland, temperate desert, multi-tree grassland, and other land use accounted for 64.30%, 16.25%, 10.37%, and 6.91% of the grassland transfer area, respectively. Multi-tree grassland decreased in area by 3014 km$^2$ and was mainly transferred to grassland and multi-tree grassland, accounting for 61.31% and 24.72%, respectively. The land use type with the largest transferred area was other land use, with an increase of 8917 km$^2$; it was mainly transferred from bare or low-coverage grassland and grassland, accounting for 73.16% and 24.61%, respectively. Temperate shrub desert was mainly transferred from grassland and multi-tree grassland, accounting for 70.11% and 21.25% of the transferred area, respectively. The transferred area of deciduous broad-leaved forest and closed shrub was greater than the un-transferred area, and the areas of evergreen broad-leaved forest and closed shrub showed small changes in area.

*3.2. Spatial and Temporal Characteristics of Future Land Change in Qinghai–Tibet Plateau under Multi-Scenario Mode*

In this study, the PLUS model was adopted to predict and analyze the spatio-temporal distribution pattern of future land use data in the Qinghai–Tibet Plateau under three scenarios (a natural development scenario, a cultivated land protection scenario, and an ecological protection scenario). The spatial distributions of land use in 2030, 2040, 2050, and 2060 are shown in Figure 4. In this paper, the land use in 2020 was utilized to validate the applicability of the PLUS model with a Kappa coefficient of 94.1%, which indicated that the PLUS model had better applicability for predicting future land use in the Qinghai–Tibet Plateau.

(1)　Natural development scenarios

As shown in Table 3, the structure of land use types in the four periods from 2030 to 2060 changes to varying degrees. Among these land use types, grassland accounts for the highest proportion, representing 52.60%, 53.01%, 53.28%, and 53.54% of the study area, respectively, and showing a trend of gradual growth. This is followed by bare or low-coverage grassland, accounting for 36.99%, 36.36%, 35.82%, and 35.28% of the study area, respectively, but showing a decreasing trend.

**Table 3.** Changes in land use structure in the Qinghai–Tibet Plateau from 2030 to 2060 under the natural development scenario.

| Land Use Type | 2030 | | 2040 | | 2050 | | 2060 | |
|---|---|---|---|---|---|---|---|---|
| | Area/$10^4$ km$^2$ | Proportion/% | Area/$10^4$ km$^2$ | Proportion/% | Area/$10^4$ km$^2$ | Proportion/% | Area/$10^4$ km$^2$ | Proportion/% |
| Coniferous forest | 3.77 | 1.49 | 3.83 | 1.51 | 3.84 | 1.51 | 3.85 | 1.52 |
| Evergreen broad-leaved forest | 0.27 | 0.11 | 0.26 | 0.10 | 0.26 | 0.10 | 0.26 | 0.10 |
| Deciduous broad-leaved forest | 0.59 | 0.23 | 0.57 | 0.22 | 0.56 | 0.22 | 0.56 | 0.22 |
| Mixed forest | 4.61 | 1.81 | 4.59 | 1.81 | 4.62 | 1.82 | 4.65 | 1.83 |
| Closed shrub | 0.09 | 0.00 | 0.09 | 0.00 | 0.09 | 0.00 | 0.09 | 0.00 |
| Temperate shrub desert | 2.46 | 0.97 | 2.44 | 0.96 | 2.41 | 0.95 | 2.40 | 0.94 |
| Multi-tree grassland | 6.22 | 2.44 | 6.15 | 2.42 | 6.11 | 2.41 | 6.05 | 2.38 |
| Grassland | 133.52 | 52.60 | 134.58 | 53.01 | 135.25 | 53.28 | 135.91 | 53.54 |
| Bare or low-coverage grassland | 93.91 | 36.99 | 92.31 | 36.36 | 90.94 | 35.82 | 89.56 | 35.28 |
| Other land use | 8.52 | 3.36 | 9.12 | 3.59 | 9.87 | 3.89 | 10.63 | 4.19 |
| Total | 253.86 | 100.00 | 2,538,635 | 100.00 | 253.86 | 100.00 | 253.86 | 100.00 |

(2)　Farmland protection scenario

Table 4 shows that the structure of land use types changes to varying degrees from 2030 to 2060 under the cultivated land protection scenario. Among them, grassland represents the highest proportion of land types, accounting for 52.60%, 53.02%, 53.28%, and 53.54%, respectively, showing a trend of gradual growth. This is followed by bare or low-coverage grassland, accounting for 36.99%, 36.38%, 35.86%, and 35.34%, respectively, showing a gradual downward trend. The proportion of other land use was 3.36%, 3.57%, 3.85%, and 4.12%, respectively, also showing a trend of gradual growth.

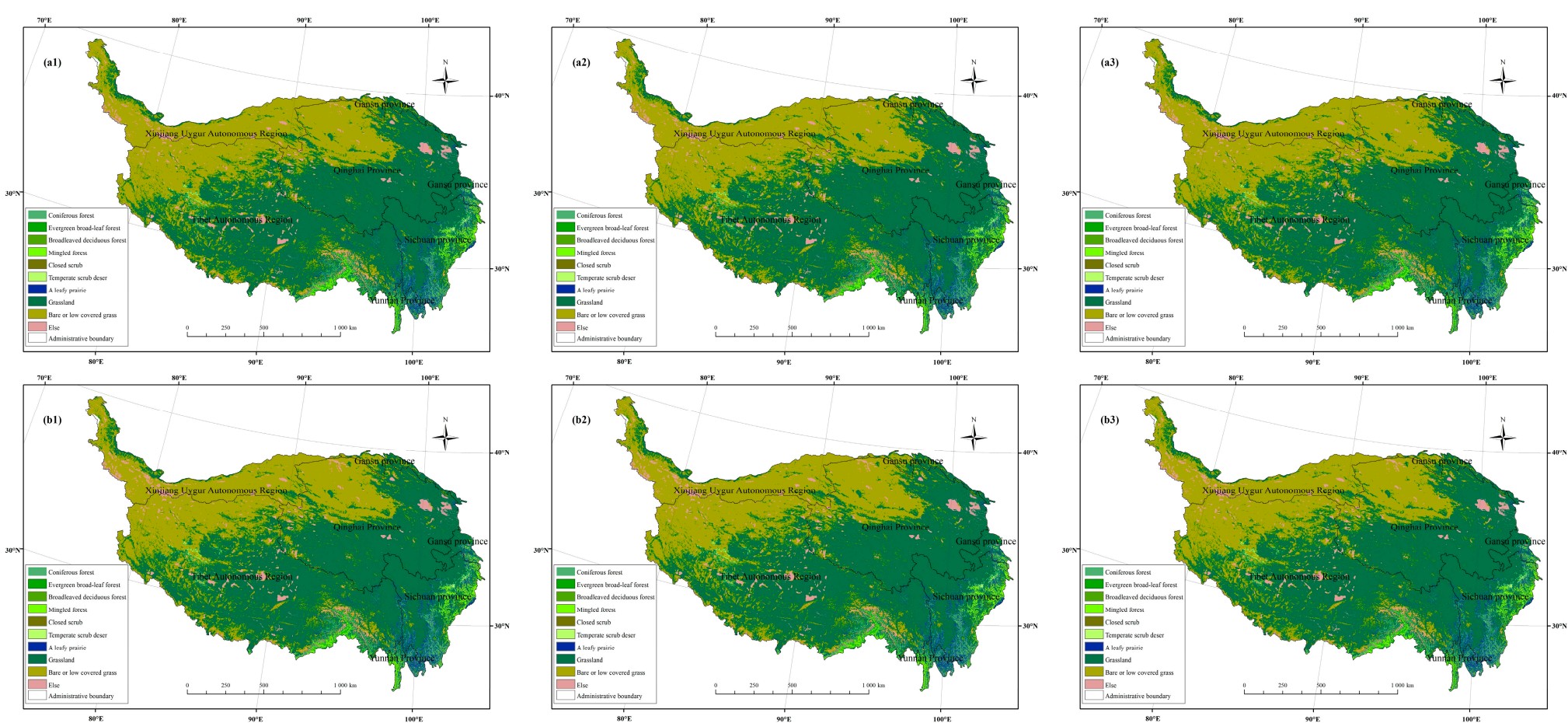

**Figure 4.** *Cont.*

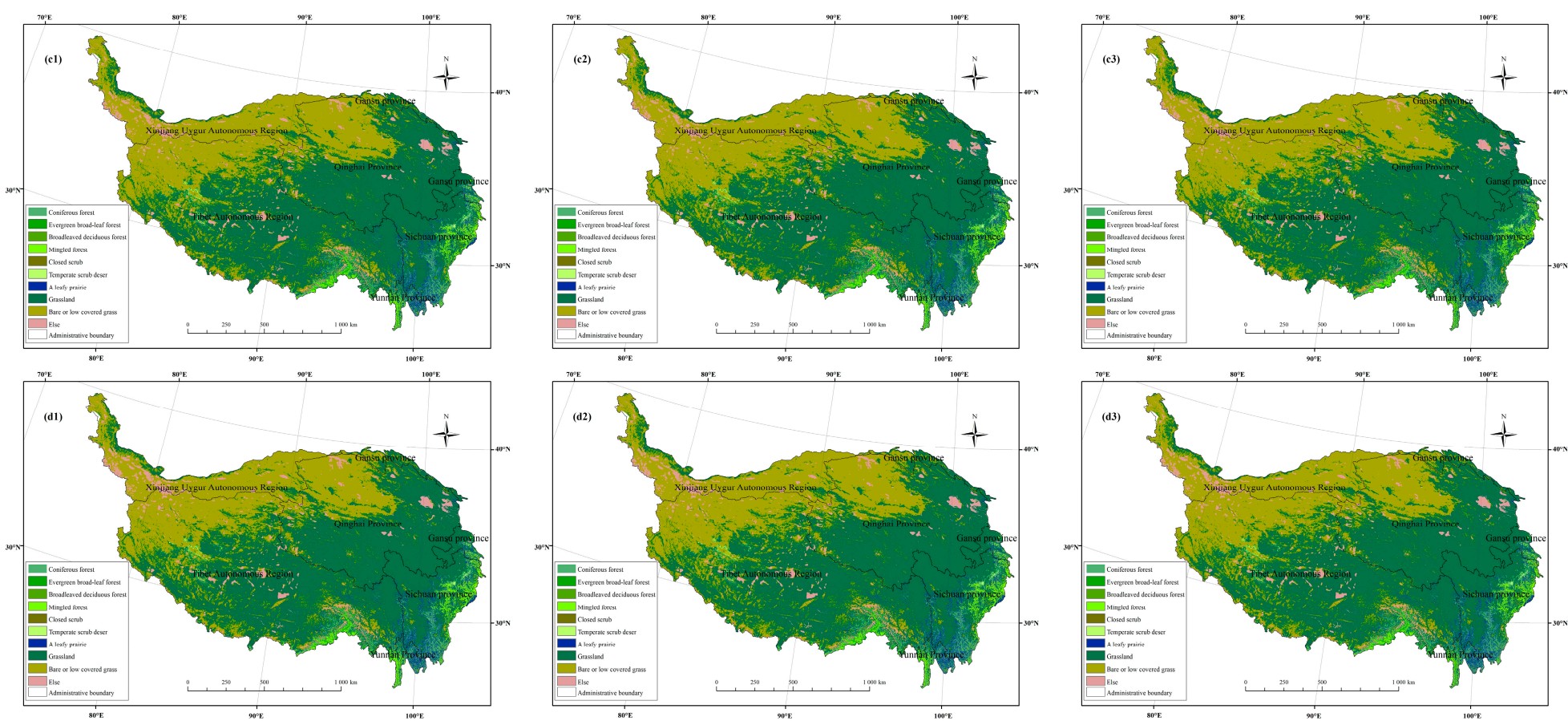

**Figure 4.** Land use simulation data for 2030~2060: natural development scenario land use simulation map for (**a1**) 2030, (**b1**) 2040, (**c1**) 2050, and (**d1**) 2060; cultivated land protection scenario land use simulation map for (**a2**) 2030, (**b2**) 2040, (**c2**) 2050, and (**d2**) 2060; and ecological protection scenario land use simulation map for (**a3**) 2030, (**b3**) 2040, (**c3**) 2050, and (**d3**) 2060.

**Table 4.** Changes in land use structure in the Qinghai–Tibet Plateau from 2030 to 2060 under the cultivated land protection scenario.

| Land Use Type | 2030 | | 2040 | | 2050 | | 2060 | |
|---|---|---|---|---|---|---|---|---|
| | Area/$10^4$ km$^2$ | Proportion/ % | Area/$10^4$ km$^2$ | Proportion/ % | Area/$10^4$ km$^2$ | Proportion/ % | Area/$10^4$ km$^2$ | Proportion/ % |
| Coniferous forest | 3.77 | 1.48 | 3.84 | 1.51 | 3.84 | 1.51 | 3.85 | 1.52 |
| Evergreen broad-leaved forest | 0.27 | 0.11 | 0.26 | 0.10 | 0.26 | 0.10 | 0.26 | 0.10 |
| Deciduous broad-leaved forest | 0.59 | 0.23 | 0.56 | 0.22 | 0.56 | 0.22 | 0.56 | 0.22 |
| Mixed forest | 4.61 | 1.81 | 4.60 | 1.81 | 4.631 | 1.82 | 4.64 | 1.83 |
| Closed shrub | 0.09 | 0.00 | 0.09 | 0.00 | 0.09 | 0.00 | 0.09 | 0.00 |
| Temperate shrub desert | 2.47 | 0.97 | 2.43 | 0.96 | 2.41 | 0.95 | 2.39 | 0.94 |
| Multi-tree grassland | 6.20 | 2.44 | 6.15 | 2.42 | 6.10 | 2.40 | 6.06 | 2.39 |
| Grassland | 133.54 | 52.60 | 134.60 | 53.02 | 135.25 | 53.28 | 135.92 | 53.54 |
| Bare or low-coverage grassland | 93.90 | 36.99 | 92.35 | 36.38 | 91.04 | 35.86 | 89.72 | 35.34 |
| Other land use | 8.52 | 3.36 | 9.07 | 3.57 | 9.77 | 3.85 | 10.46 | 4.12 |
| Total | 253.86 | 100.00 | 253.86 | 100.00 | 253.86 | 100.00 | 253.86 | 100.00 |

(3)　Ecological protection scenario

As shown in Table 5, grassland is the most widely distributed land type from 2030 to 2060, accounting for 52.60%, 53.01%, 53.28%, and 53.54%, respectively, showing a trend of gradual increase (increasing by 0.94%). This is followed by bare or low-coverage grassland, accounting for 36.99%, 36.38%, 35.86%, and 35.34%, respectively, showing a trend of gradual decrease (decreased by 1.65%). Compared with the natural development scenario and the farmland protection scenario, the areas of forest, evergreen broad-leaved forest, and deciduous broad-leaved forest had larger changes in land type structure. The proportion of forest area in the four periods was 1.51%, 1.56%, 1.60%, and 1.62%, respectively. The proportion of evergreen broad-leaved forest area was 0.11%, 0.11%, 0.11%, and 0.11%, respectively, while that of deciduous broad-leaved forest was 0.26%, 0.28%, 0.30%, and 0.30%, respectively.

**Table 5.** Changes in land use structure in the Qinghai–Tibet Plateau from 2030 to 2060 under the ecological protection scenario.

| Land Use Type | 2030 | | 2040 | | 2050 | | 2060 | |
|---|---|---|---|---|---|---|---|---|
| | Area/$10^4$ km$^2$ | Proportion/% | Area/$10^4$ km$^2$ | Proportion/% | Area/$10^4$ km$^2$ | Proportion/% | Area/$10^4$ km$^2$ | Proportion/% |
| Coniferous forest | 3.84 | 1.51 | 3.97 | 1.56 | 4.07 | 1.60 | 4.10 | 1.62 |
| Evergreen broad-leaved forest | 0.27 | 0.11 | 0.27 | 0.11 | 0.27 | 0.11 | 0.27 | 0.11 |
| Deciduous broad-leaved forest | 0.65 | 0.26 | 0.70 | 0.28 | 0.76 | 0.30 | 0.76 | 0.30 |
| Mixed forest | 4.64 | 1.83 | 4.31 | 1.70 | 4.18 | 1.65 | 4.17 | 1.64 |
| Closed shrub | 0.08 | 0.00 | 0.09 | 0.00 | 0.08 | 0.00 | 0.08 | 0.00 |
| Temperate shrub desert | 2.52 | 0.99 | 2.43 | 0.96 | 2.40 | 0.95 | 2.38 | 0.94 |
| Multi-tree grassland | 5.97 | 2.35 | 6.15 | 2.42 | 6.11 | 2.40 | 6.05 | 2.38 |
| Grassland | 133.54 | 52.60 | 134.58 | 53.01 | 135.25 | 53.28 | 135.91 | 53.54 |
| Bare or low-coverage grassland | 93.91 | 36.99 | 92.36 | 36.38 | 91.04 | 35.86 | 89.73 | 35.34 |
| Other land use | 8.52 | 3.36 | 9.07 | 3.57 | 9.77 | 3.85 | 10.46 | 4.12 |
| Total | 253.86 | 100.00 | 253.86 | 100.00 | 253.86 | 100.00 | 253.86 | 100.00 |

### 3.3. Spatial and Temporal Evolution of Carbon Storage in Qinghai–Tibet Plateau from 2030 to 2060 under Multi-Scenario Model

Carbon storage was calculated using the InVEST model for 2020 with an overall accuracy of 91.3%; this was validated with the field observation data obtained by the Institute of Geographic Sciences and Natural Resources Research, CAS. The applied models had better applicability in the study region.

Under the natural development scenario, the carbon storage in the Qinghai–Tibet Plateau in 2030, 2040, 2050, and 2060 was calculated to be $434.00 \times 10^9$ Mg, $432.19 \times 10^9$ Mg, $430.49 \times 10^9$ Mg, and $428.80 \times 10^9$ Mg, respectively, showing a trend of 'gradual decrease' (decreasing by $5.2 \times 10^9$ Mg). From 2030 to 2060, the total stored carbon decreases by $520.00 \times 10^7$ Mg, with an overall decrease of 1.21%. The amount of stored carbon changes by $-181.00 \times 10^7$ Mg from 2030 to 2040, $-170.00 \times 10^7$ Mg from 2040 to 2050, and $-169.00 \times 10^7$ Mg from 2050 to 2060.

Under the cultivated land protection scenario, the carbon storage in the Qinghai–Tibet Plateau in 2030, 2040, 2050, and 2060 was calculated to be $434.03 \times 10^9$ Mg, $432.28 \times 10^9$ Mg, $430.71 \times 10^9$ Mg, and $429.06 \times 10^9$ Mg, respectively, showing a trend of 'gradual decrease' (decreasing by $4.97 \times 10^9$ Mg). From 2030 to 2060, the total stored carbon decreases by $497.00 \times 10^7$ Mg, with an overall decrease of 1.14%. The amount of stored carbon changes by $-175.00 \times 10^7$ Mg (2030~2040), $-157.00 \times 10^7$ Mg (2040~2050), and $-165.00 \times 10^7$ Mg (2050~2060).

Under the ecological protection scenario, the carbon storage in the Qinghai–Tibet Plateau in 2030, 2040, 2050, and 2060 was calculated to be $434.38 \times 10^9$ Mg, $430.99 \times 10^9$ Mg, $428.68 \times 10^9$ Mg, and $426.96 \times 10^9$ Mg, respectively, showing a trend of 'gradual decrease' (decreasing by $7.42 \times 10^9$ Mg). From 2030 to 2060, the total stored carbon decreases by $742.00 \times 10^7$ Mg, accounting for 1.70%. The amount of stored carbon changes by $-339.00 \times 10^7$ Mg (2030~2040), $-231.00 \times 10^7$ Mg (2040~2050), and $-172.00 \times 10^7$ Mg (2050~2060).

The carbon pool had significant influence on the carbon dioxide content in the atmosphere and can be divided into two types: carbon source and carbon sink [24]. The carbon storage difference (carbon sequestration value) was calculated for the Qinghai–Tibet Plateau: an area with a higher absolute value than the positive value (>500 Mg) was reclassified as a carbon sink area, an area with a value around 0 ($-500$~500 Mg) was reclassified as a carbon balance area, and an area with a higher absolute value than the negative value ($<-500$ Mg) was reclassified as a carbon source area (Figure 5).

Under the natural development scenario, the vast majority of the Qinghai–Tibet Plateau is represented by carbon balance areas, and the carbon source and carbon sink areas are scattered in various prefectures. From 2030 to 2040, the carbon source areas are mainly distributed in Kashgar City, Hotan City, and Wulan County, while the carbon sink areas are mainly located in Jiuquan City and the southern and southeastern edges of the Qinghai–Tibet Plateau. From 2040 to 2050, the carbon source areas expand further, and are mainly distributed in Kashgar City, Hotan City, northern Gar County, Wulan County, and Jiuquan City, while the carbon sink areas are mainly distributed in the Zhiduo County, Korla City, and the Jiuquan City and Wulan County boundary area. From 2050 to 2060, compared with the previous periods, the carbon source areas and carbon sink areas continue to expand. The carbon source areas are mainly distributed in the south of Atushi City, Kashgar City, Hotan City, northern Gar County, and Wulan County, while the carbon sink areas are concentrated in the border between Korla City and Zhiduo County, Jiuquan City, and Wulan County.

Under the cultivated land protection scenario, most of the Qinghai–Tibet Plateau is represented by carbon balance areas, and the carbon source and carbon sink areas are scattered in various prefectures. From 2030 to 2040, the carbon source areas are mainly distributed in Kashgar City, Hotan City, and northern Wulan County, while the carbon sink areas are mainly distributed in Kashgar City, Jiuquan City, and other areas. From 2040 to 2050, the carbon source and carbon sink areas are further expanded. The carbon source

areas are mainly distributed in Kashgar City, Hotan City, northern Gar County, and the Wulan County and Jiuquan City boundary zone, while the carbon sink areas are mainly distributed in the Korla City and Zhiduo County boundary and the Wulanshi and Jiuquan City boundary. From 2050 to 2060, compared with the previous period, the carbon source and carbon sink areas continue to expand. The carbon source areas are mainly distributed in Kashgar, Hotan City, northern Gar County, Wulan County, and other regions, while the carbon sink areas are mainly distributed in the boundary between Wulan County and Zhiduo County, Jiuquan City, and Wulan County.

Under the ecological protection scenario, most of the Qinghai–Tibet Plateau is represented by carbon balance areas, and the carbon source and carbon sink areas are scattered in various prefectures. From 2030 to 2040, the carbon source areas are mainly located in Kashgar City, Hotan City, and the northern part of Wulan County, while the carbon sink areas are mainly located in the southern part of Atushi City and the southern part of Kashgar City. From 2040 to 2050, compared with the previous periods, the carbon source and carbon sink areas are further expanded. The carbon source areas are mainly distributed in Kashgar City, Hotan City, northern Wulan County, Jiuquan City, Wulan County, and Zhiduo County, while the carbon sink areas are mainly concentrated in the boundary between Jiuquan City and Wulan County and the boundary between Wulan County, Zhiduo County, and Korla City. From 2050 to 2060, compared with the previous periods, the carbon source and carbon sink areas are further expanded. The carbon source areas are mainly concentrated in Wulan County, Jiuquan City, Kashgar City, Hotan City, northern Gar County, and northwestern Zhiduo County. The carbon sink areas are concentrated on the boundary between Jiuquan City and Wulan County and the boundary between Wulan County, Zhiduo County, and Korla City.

*3.4. Carbon Storage Change Caused by Land Use Conversion*

From 2000 to 2020, the total carbon storage decreased by $380.51 \times 10^7$ Mg. The total areas of coniferous forest and evergreen broad-leaved forest increased by 2568 km$^2$ and 25 km$^2$, which resulted in the carbon storage increasing by $179.48 \times 10^6$ Mg and $976.60 \times 10^4$ Mg, respectively. The areas of deciduous broad-leaved forest and mixed forest decreased by 1687 km$^2$ and 470 km$^2$, while the carbon storage decreased by $134.04 \times 10^6$ Mg and $253.79 \times 10^6$ Mg, respectively. The areas of closed shrub, decreased temperate shrub desert, multi-tree grassland, grassland, and other land use increased by 7 km$^2$, 9297 km$^2$, 2737 km$^2$, $152.00 \times 10^2$, and 9481 km$^2$, which led to increases in carbon storage of $387.72 \times 10^4$ Mg, $175.86 \times 10^7$ Mg, $638.30 \times 10^5$ Mg, $224.81 \times 10^7$ Mg, and $202.81 \times 10^6$ Mg. The area of bare or low-coverage grassland decreased by $371.58 \times 10^2$ km$^2$, and the carbon storage decreased by $789.53 \times 10^7$ Mg.

Under the natural development scenario, the total carbon storage would decrease by $519.47 \times 10^7$ Mg from 2030 to 2060. The area of forest would increase by 780 km$^2$, leading to an increase in carbon storage of $545.16 \times 10^5$ Mg. The areas of evergreen broad-leaved forest and deciduous broad-leaved forest would decrease by 118 km$^2$ and 347 km$^2$, leading to decreases in carbon storage of $460.95 \times 10^5$ Mg and $275.71 \times 10^5$ Mg, respectively. The areas of mixed forest, grassland, bare or low-cover grassland, and other land use would increase by 514 km$^2$, $237.27 \times 10^2$ km$^2$, 4351 km$^2$, and $210.89 \times 10^2$ km$^2$, resulting in increases in carbon storage of $277.55 \times 10^6$ Mg, $350.92 \times 10^7$ M, $924.54 \times 10^7$, and $451.11 \times 10^6$ Mg, respectively. The areas of closed shrub, temperate shrub desert, and multi-tree grassland would decrease by 1 km$^2$, 710 km$^2$, and 1422 km$^2$, leading to decreases in carbon storage of $553.89 \times 10^3$ Mg, $134.30 \times 10^6$ Mg, and $331.62 \times 10^5$ Mg, respectively.

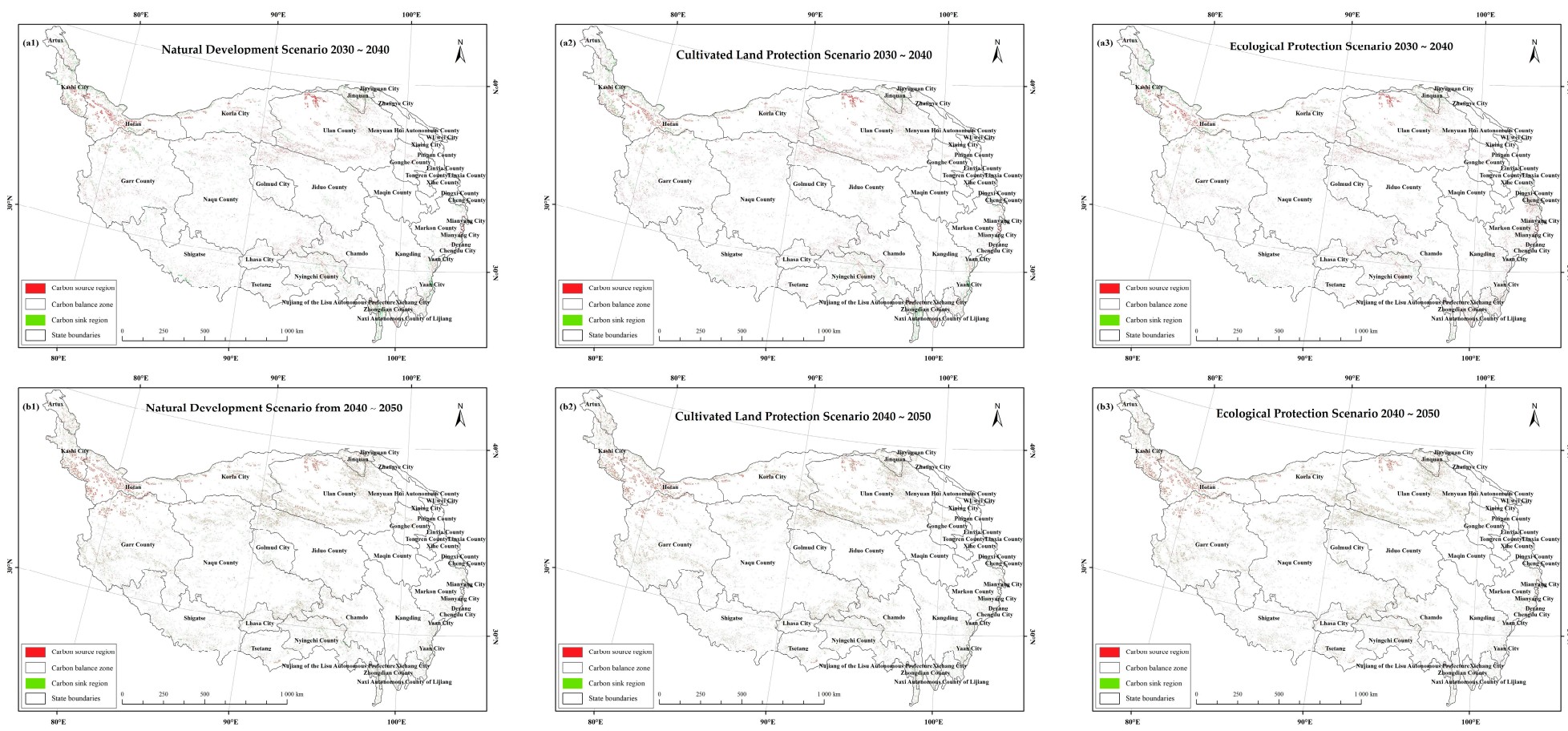

**Figure 5.** *Cont.*

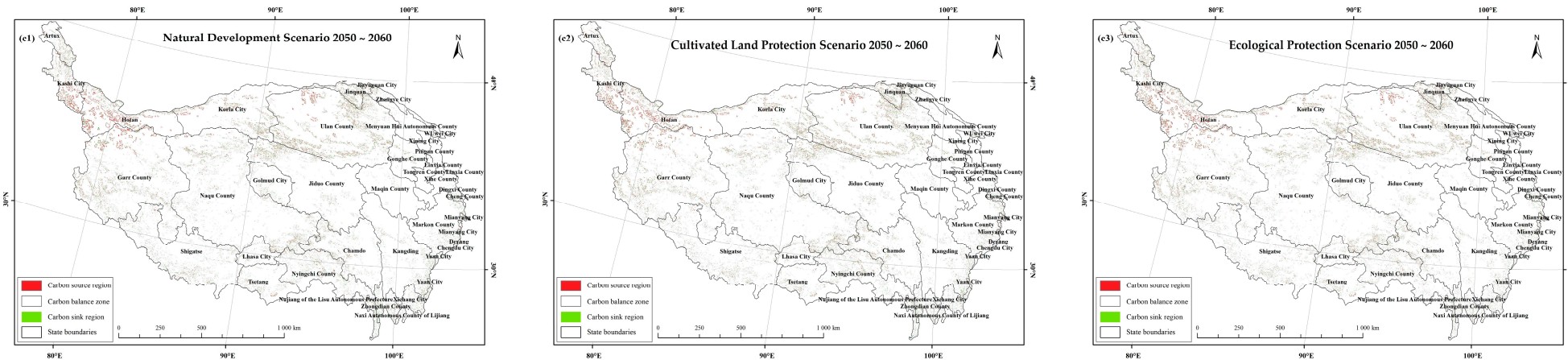

**Figure 5.** Spatial distribution of carbon source, carbon sink, and carbon balance areas.

Under the cultivated land protection scenario, the total carbon storage would decrease by $496.26 \times 10^7$ Mg from 2030 to 2060. The areas of arboreal forest, mixed forest, grassland, and other land use would increase by 825 km$^2$, 336 km$^2$, $237.79 \times 10^2$ km$^2$, and $194.66 \times 10^2$ km$^2$, leading to increases in carbon storage of $576.61 \times 10^5$ Mg, $181.44 \times 10^6$ Mg, $351.69 \times 10^7$ Mg, and $416.40 \times 10^6$ Mg, respectively. The areas of evergreen broad-leaved forest, deciduous broad-leaved forest, closed shrub, temperate shrub desert, multi-tree grassland decreased, and bare or low-coverage grassland would decrease by 108 km$^2$, 299 km$^2$, 1 km$^2$, 801 km$^2$, 1382 km$^2$, and $418.15 \times 10^2$ km$^2$, resulting in decreases in carbon storage of $421.89 \times 10^5$ Mg, $237.57 \times 10^5$ Mg, $553.89 \times 10^3$ Mg, $151.52 \times 10^6$ Mg, $322.30 \times 10^5$ Mg, and $888.48 \times 10^7$ Mg, respectively.

Under the ecological protection scenario, the total carbon storage would decrease by $742.19 \times 10^7$ Mg from 2030 to 2060. The areas of forest, evergreen broad-leaved forest, deciduous broad-leaved forest, closed shrub, multi-tree grassland, grassland, and other land use would increase by 2611 km$^2$, 1 km$^2$, 1142 km$^2$, 1 km$^2$, 862 km$^2$, $237.83 \times 10^2$ km$^2$, and $194.60 \times 10^2$ km$^2$, leading to increases in carbon storage of $182.49 \times 10^6$ Mg, $390.63 \times 10^3$ Mg, $907.39 \times 10^5$ Mg, $553.89 \times 10^3$ Mg, $201.03 \times 10^5$ Mg, $351.75 \times 10^7$ Mg, and $416.27 \times 10^6$ Mg, respectively. The areas of mixed forest, temperate shrub, and bare or low-coverage grassland would decrease by 4622 km$^2$, 1416 km$^2$, and $418.22 \times 10^2$ km$^2$, resulting in decreases in carbon storage of $249.58 \times 10^7$ Mg, $267.85 \times 10^6$ Mg, and $888.63 \times 10^7$ Mg (Figure 6).

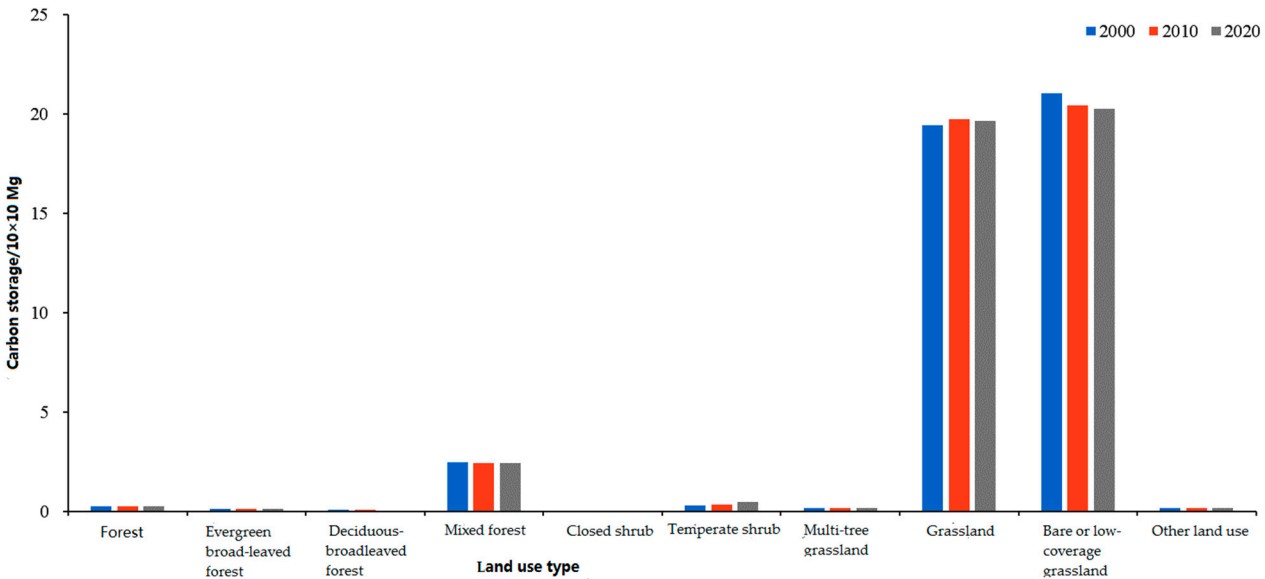

**Figure 6.** Carbon storage in different land use types in 2000, 2010, and 2020.

### 3.5. Dominant Factors of Carbon Storage Changes in Qinghai–Tibet Plateau

In this study, the temperature, precipitation, population, GDP, elevation, slope, distance from railways, distance from national highways, and distance from provincial highways during 2000–2020 were used as the X independent variables, and the carbon storage in the Qinghai–Tibet Plateau was used as the Y dependent variable. Geodetector was used to analyze and detect the explanatory power of X for the spatial differentiation of Y.

Figure 7 showed that the q values of the nine driving factors affecting the spatial and temporal changes in carbon storage in the Qinghai–Tibet Plateau were ordered as follows: precipitation > elevation > temperature > distance from provincial highway > distance from railway > slope > GDP > distance from national highway > population. Among these, the maximum q value was for precipitation, which was 0.116, followed by elevation, with a q value of 0.038; the q value of the distance from provincial highways was 0.029, and the q value of the distance from railways was 0.011.

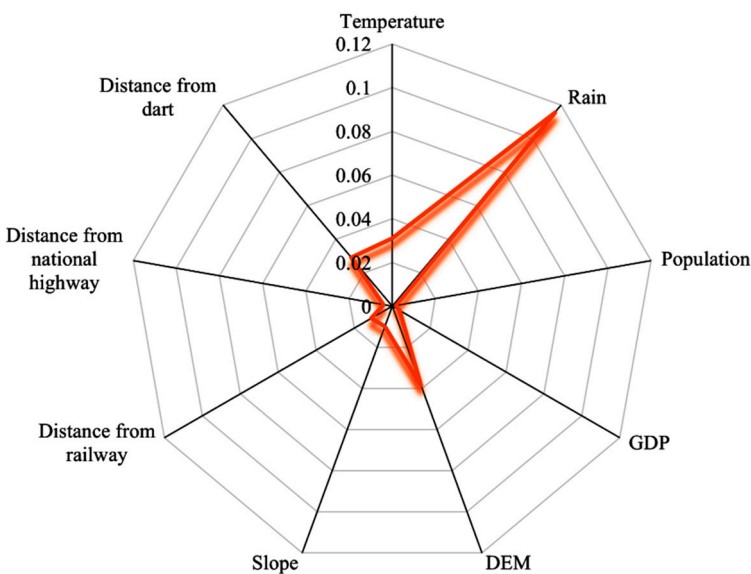

**Figure 7.** Explanatory power of each single factor (q value).

Figure 8 shows that the q value of the interaction of driving factors was greater than the single factor q value (diagonal value), indicating that the driving factors did not influence the change in carbon storage independently, but rather that there was a process of interaction and mutual enhancement. The interaction between the two driving factors during 2000–2020 was mainly dominated by nonlinear enhancement. The four groups with higher explanatory power for carbon storage changes were as follows: precipitation ∩ elevation (0.888) > precipitation ∩ slope (0.696) > precipitation ∩ temperature (0.663) > precipitation ∩ distance from the railway (0.642).

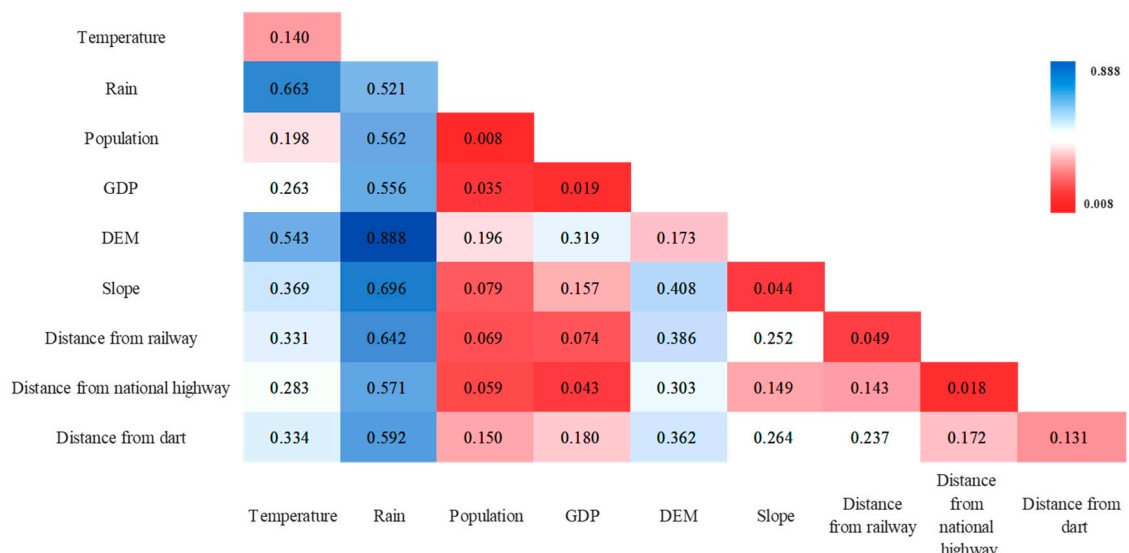

**Figure 8.** The q value of each interaction factor.

## 4. Discussion

It was found in this study that the Qinghai–Tibet Plateau was mainly covered by grassland and bare or low-coverage grassland, while the areas covered by multi-tree grassland, mixed forest, coniferous forest, deciduous broad-leaved forest, and evergreen broad-leaved forest were relatively smaller. Of these land use types, grassland was mainly distributed in the central and southern regions, while bare or low-coverage grassland was mainly located in the northern region, which is related to the higher altitude, scarce precipitation,

and lower temperature in this region [25]. The area of multi-tree grassland was mainly distributed in the southeastern region, while mixed forest was mainly distributed in the southern edge and southeastern region. The reason for this was that these regions are characterized by a higher temperature, abundant precipitation, and relatively low altitude, which are conductive to vegetation growth [26–28]. The temperate shrub desert was mainly distributed in the central part of Maerkang County, the southern part of Gar County, and the southern edge of the study area, where the hydrothermal conditions are relatively poor.

The total carbon storage in the Qinghai–Tibet Plateau under the three scenarios for 2000~2020 and 2030~2060 showed a gradual decreasing trend. Carbon storage decreases the most under the ecological protection scenario from 2030 to 2060. This is mainly due to the fact that in the west and northwest parts of the Qinghai–Tibet Plateau, desert and bare or low-vegetation zones are widely distributed. Under the ecological protection scenario, the decreasing trend was the most obvious because the land use types with a lower capacity for carbon storage in these above regions were not allowed to change [29]. The carbon storage decreased the least in the natural development scenario. In 2030, under different scenarios, the ranking of total carbon storage in the Qinghai–Tibet Plateau is as follows: ecological protection scenario, cultivated land protection scenario, natural development scenario. The ranking for 2040–2060 is as follows: cultivated land protection scenario, natural development scenario, and ecological protection scenario. This is due to the fact that with the application of a cultivated land protection policy, more farmland with a greater carbon sequestration capacity would be cultivated in zones with lower vegetation coverage [30]. Grassland, bare or low-coverage grassland, and mixed forest are the main land types with high carbon storage in the three scenarios from 2030 to 2060. In the three scenarios from 2000 to 2060, the vast majority of regions showed carbon balance areas, while the carbon source and carbon sink areas were scattered over the whole study region. The highest carbon storage in the three scenarios from 2000 to 2060 was $553.89 \times 10^2$ Mg, while the lowest carbon storage was $291.39 \times 10^2$ Mg. Zones with higher carbon storage were mainly distributed in the southeast of Zedang County, the south of Linzhi County, Nujiang Lisu Autonomous Prefecture, and the south of Maerkang County in the north of the Qinghai–Tibet Plateau. The reason for this lies in the fact that in these regions, forests and shrubs with a higher capacity for carbon storage are widely distributed. The lowest carbon storage areas are scattered in the west and north parts of the study area, including Hotan City, Naqu County, Gar County, Wulan County, Xining City, Gonghe County, and Menyuan Hui Autonomous County. The reason for this is that in the above regions, precipitation is scarce and the temperature is lower, so desert and lower-vegetation zones with a lower capacity for carbon storage are widely distributed.

During 2000–2020, it was found that the zones in the study area with no vegetation expanded rapidly from 2000 to 2020, occupying a large area of low-coverage grassland, grassland, montane coniferous forests, and dwarf grass [31,32]. The reason for this is that grassland and forest were destroyed and more built-up areas were developed due to Western development and urbanization processes [33]. Meanwhile, during 2030–2060, the cultivated land protection scenario of ecological priority development scenario could effectively promote the increase in deciduous shrub, closed shrub, dwarf grass, grassland, and low-coverage grassland area. In view of these phenomena, the following suggestions for regional land use planning in the Qinghai–Tibet Plateau were proposed:

(1) Reasonable planning of urban construction and development boundaries, promoting the coordinated development of urban and rural areas. The prediction results of vegetation types under different scenario constraints prove that urban development needs to be controlled and guided. On the one hand, it is necessary to control the growth of construction land areas, manage the transfer of land, and strengthen the linkage supervision of construction land. On the other hand, the boundary of urban expansion should be reasonably planned to ensure a centralized connection and a reasonable shape [34].

(2) Establish the concept of an ecological red line, achieving ecological co-governance and environmental co-protection. The ecological protection red line is an important control boundary in territorial space planning. It plays an important role in promoting the balanced development of the population, resources, and the environment, and the complementarity and coordination of economic, social, and ecological benefits [35]. Developmental and productive construction activities are prohibited in ecological protection areas, and close attention is paid to occupied ecological protection areas under inertial development scenarios [36].

## 5. Conclusions

In this study, the spatial and temporal evolution patterns of carbon storage in the Qinghai–Tibet Plateau were explored, and then the dominant driving factors in different periods and scenarios were determined. The main conclusions were as follows:

(1) During 2000–2020, grassland and bare or low-coverage grassland were the main land use types in the Qinghai–Tibet Plateau. They were mainly distributed in the central, southern, and northern parts.

(2) During 2000–2020, the areas of coniferous forest, evergreen broad-leaved forest, closed shrub, temperate desert shrub, multi-tree grassland, and grassland increased, while the areas of deciduous broad-leaved forest, mixed forest, and bare or low-coverage grassland decreased.

(3) During 2030–2060, it was found that the total carbon storage in the Qinghai–Tibet Plateau under three different development scenarios would gradually decrease as a result of a transformation of grassland to non-vegetation zones.

(4) During 2000–2020, the dominant factor affecting the changes in carbon storage in the Qinghai–Tibet Plateau was precipitation, followed by topographic factors.

(5) In future ecological protection and restoration efforts, more high-quality farmlands should be protected and constructed. This could contribute to the achievement of dual-carbon goals. In addition, hydrothermal conditions should be improved to enhance the carbon cycle process in the Qinghai–Tibet Plateau.

Under the context of global change, the dominant driving factors in different periods were different. In addition, due to the geospatial differentiation of the Qinghai–Tibet Plateau, the spatio-temporal change patterns and their driving mechanisms differed greatly. Further studies should be carried out to investigate the above problems.

**Author Contributions:** Conceptualization, methodology, and writing—original draft preparation: H.Z. and C.Y.; investigation, supervision, project administration, and funding acquisition: M.L. and L.W.; methodology and investigation: B.G. All authors have read and agreed to the published version of the manuscript.

**Funding:** This research was funded by the National Natural Science Foundation of China (grant numbers: 52101405, 42101306, 42071419 and 62006096); the Natural Science Foundation of Shandong Province (grant number: ZR2021MD047); and the Scientific Innovation Project for Young Scientists in Shandong Provincial Universities (grant number: 2022KJ224), and Agricultural Science and Technology Innovation Program (grant number CAAS-ZDRW202201).

**Data Availability Statement:** The data presented in this study are available on request from the corresponding author. The data are not publicly available due to the data privacy.

**Conflicts of Interest:** The authors declare no conflicts of interest. Caifeng Yang is employed by Beijing WANYUN Science & Technology Development Co., Ltd., and she is a graduate student who has already graduated from our team. However, she contributed to this paper (including the investigation and original paper writing) when she was in our team. However, her company did not participate in the research experiment or manuscript writing.

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
