# Peer review of "Patterns and Dominant Driving Factors of Carbon Storage Changes in the Qinghai–Tibet Plateau under Multiple Land Use Change Scenarios"

_forests, doi:10.3390/f15030418_

Round 1
Reviewer 1 Report
Comments and Suggestions for Authors
1) In the introduction section the author stated numerous literature on carbon dynamics with land use changes and climate change. However, I don’t found a sound justification for this study. The author should state that how this study is different from other.
2) Study area needed sources of citation
3) Can the author calculate the annual rate of land use change dynamics please refer to Ahmad, A., Liu, Q. I. J., Nizami, S. M., Mannan, A., Saeed, S. (2018): Carbon emission from deforestation, forest degradation and wood harvest in the temperate region of hindukush himalaya, pakistan between 1994 and 2016. – Land Use Policy 78: 781-790. DOI: https://doi.org/10.1016/j.landusepol.2018.07.009
4) Line 203-207, Please make it simple. Difficult to comprehend
5) Line 2018-233. Please make it simple. Need rephrasing
6) Please check line 283
7) Among them, grassland accounted for the highest proportion, accounting for 52.60%, 53.01%, 53.28% and 53.54’’ Among them, grass- land was the highest proportion of land types, accounting for 52.60%, 53.02%, 53.28% and 271 53.54%’’ grassland was the most widely distributed land type, accounting for 52.60%, 53.01%, 53.28% and 53.54% respectively” In all scenario grassland is increasing with very little variation, so what is the scope of this.
8) Line 295-297. Repetition
9) Line 300-301: From 2030 to 2060, the total carbon storage decreased by 519.47×107 Mg. Confusing , how this occurred
10) The carbon sequestration was -180.76×107 Mg from 2030 to 2040, -169.59×107 Mg from 2040 to 2050, and -169.11×107 Mg from 2050 to 2060. Why the values are negative
11) Line 298-315: Very confusing results and difficult to pick. Particularly the decrease value
12) Line 310-315: Sequestration value is too much high compared to stock
13) Line 316-322: What is the scope of this here
14) Line 464-467: How ecological protection policies decrease carbon?
15) The discussion section is very weak and need major improvement
16) Conclusion section need much improvement. Please include implication and application of this study
17) Please go through the following articles
Jia, B., & Zhou, G. (2023). Estimation of global karst carbon sink from 1950s to 2050s using response surface methodology. Geo-spatial Information Science. doi: 10.1080/10095020.2023.2165974
Qiu, S., Yang, H., Zhang, S., Huang, S., Zhao, S., Xu, X.,... Banwart, S. A. (2023). Carbon storage in an arable soil combining field measurements, aggregate turnover modeling and climate scenarios. CATENA, 220, 106708. doi: https://doi.org/10.1016/j.catena.2022.106708
Jiang, C., Wang, Y., Yang, Z., & Zhao, Y. (2023). Do adaptive policy adjustments deliver ecosystem-agriculture-economy co-benefits in land degradation neutrality efforts? Evidence from southeast coast of China. Environmental Monitoring and Assessment, 195(10), 1215. doi: 10.1007/s10661-023-11821-6
1
Comments on the Quality of English LanguageThe quality of data presentation of results, discussion and conclusion Need improvement. The results are confusing need to be made simple. Discussion need to be coherent and connected
Reviewer 2 Report
Comments and Suggestions for Authors
The current investigation entitled “Change patterns and its dominant driving factors of carbon storage in the Qinghai-Tibet Plateau under multiple land use change scenarios” authored by Zhao et al. utilised the PLUS model to predict the land use change data under the natural development scenario, cultivated land protection scenario and ecological protection scenario of the Qinghai-Tibet Plateau from 2030 to 2060, and applied the InVEST model to obtain the carbon storage in different periods and different scenarios. Moreover, the spatial-temporal evolutions of carbon storage in the Qinghai-Tibet Plateau under various scenarios had been analyzed, and the driving mechanisms of carbon storage were clarified by Geodetector. The finding of the current investigation could provide decision supports for the protection of regional ecosystems and the realization of regional dual-carbon goals in the studyregion.
Comments/suggest
Abstract
Line 14-15. The meaning of the statement is not clear. Kindly revise and should indicate the background information about the current investigation.
Line 16-18. Just repetition of the statement provided in the first statement and title of the manuscript. The author should provide more constructive statement. Thus revise the statement.
The finding of the investigation should be more quantitative rather than descriptive. The Line 27-29, also need to be revised for better conclusive statement.
Thus overall modify the abstract section.
The keyword section also need to be revised and should not include the words already included in the title.
Introduction
The author should avoid to use abbreviations without explaining them such as SD provided in the Line50. So kindly make the correction throughout the manuscript.
The introduction section is well written covering all the aspect i.e., background information, previous literature and research gap. However, I suggest author to provide the specific objectives of the current investigation.
Material and methods
In subsection 2.1, the author should also provide the detail about the edaphic factors. Moreover, the author should also provide a flow chart indicating the methodology adopted in the current investigation for better clarity to the readers.
Result
Title of the subsection 3.1 need to be revised. I think it will be better if it is limited to “Spatial and temporal characteristics of land use change”
Line 184. Does it really need to start the paragraph with “As shown in Table 2”. The author can directly start the paragraph and table and figure in the brackets. The authors have not mentioned the accuracy of the land use cover used in the current investigation. The result seems good, However I suggest authors, kindly improve the quality of the figures, since some of the figures are very tough to read.
The discussion is well written, however, the author to read new literature to strengthen the discussion section. Moreover, the author also need to remove the generalised statement and should be limited to discussion of the finding of the current investigation.
The conclusion section needs to revise. First there is no need to provide the methodological detail in the conclusion section. Moreover conclusion should be limited to the constructive finding of the current investigation rather than just replica of the result section. Simultaneously, the limitations as well as the way forward also need to be indicated.
Comments on the Quality of English Language
Moderate editing of English language required
Reviewer 3 Report
Comments and Suggestions for Authors
The article provides long term modelling results for carbon stock in the Qinghai-Tibet Plateau. This information is important for the climate and land use policy and evaluation of consequences of prioritizing different management strategies. However, the article can be improved. The most significant shortcoming is insufficient presentation of validation of the applied models, which would be obvious considering that the historical land use data are elaborated for the study and the models can be validated using these data. Another methodological improvement is more detailed presentation of the climate scenario(s) used in the study.
Table 1 should be reorganized. It is now hard to follow to the text.
In the methodology it is not clearly state if the undergrowth biomass (in forest land) is considered and if litter layer is accounted with soil. I'm not sure if it is in scope, but if forests are managed, at least part of them, harvested wood products should be considered, as well as substitution effect of biofuel. This may increase difference between management and nature conservation elated scenarios.
In Figure 3 it is hard to see difference between the scenarios. It would be more informative to add diagram with land use to each map.
In the results chapter, particularly, starting from line 259, without uncertainty range these values are not informative. It is not clearly explained how the significance of difference or trend is determined. Here results of validation of the models would help to compare the scenarios, even if there would not be anymore significant difference between the scenarios. It would be good to add uncertainty also in Table 3, if possible.
I am not used to some measurement units, e.g. in line 299, why not to use Mg, Gg or Tg without multiplying. It would also be more informative to provide not only total, but also per ha carbon stock and stock changes.
In conclusion starting from line 310, is this correct that area of forests increases and grasslands with bigger biomass stock increases and at the same time total carbon stock decreases? If it is so, what are the driving forces for decrease of total carbon stock?
Figure 4 is not sufficiently informative. It is not possible to distinguish green and red spots. It would also be easier to follow to presentation if title of scenario is added to map instead of caption.
In line 383, it is hard to follow here, if carbon storage increased or decreased. If decreased, what is the reason, considering that the total forest area increased? In the same chapter, how decrease of area bare ground and low-coverage grassland could lead to decrease of carbon stock? Is it so that these areas turned into deserts? What was the source of carbon in bare ground, or the definition of bare ground does not mean area without vegetation? It would be informative to add in the methodology or somewhere else definitions of land use and average carbon stock in steady stage conditions or periodic average for different land use categories in a graph, otherwise it is complicated to follow to the logic of results.
Introduction in the conclusion is not necessary (lines 522-528). Conclusion 3 should be supplemented with the main driving forces for changes.
Round 2
Reviewer 1 Report
Comments and Suggestions for Authors
Please go through for minor language correction
Comments on the Quality of English Languageminor language and spelling correction are needed
Reviewer 2 Report
Comments and Suggestions for Authors
In the first round of the revision, the author have made considerable corrections in the manuscript. however, i suggest author some more revision in the manuscript.
It will be better if table 6 can be provided as table for better clarity to the readers.
In table 8, is it possible to provide the significance level. If yes, then it will be better if it can be provided.
In the conclusion section at the end limitations of the current investigation and way forward should be provided.
Regards